# Objectively Measured Physical Activity and Sedentary Behaviour on Cardiovascular Risk and Health-Related Quality of Life in Adults: A Systematic Review

**DOI:** 10.3390/healthcare12181866

**Published:** 2024-09-16

**Authors:** Beatriz Santos, Diogo Monteiro, Fernanda M. Silva, Gonçalo Flores, Teresa Bento, Pedro Duarte-Mendes

**Affiliations:** 1ESECS—Polytechnic of Leiria, 2411-901 Leiria, Portugal; beatrizfilipa97@gmail.com (B.S.); diogo.monteiro@ipleiria.pt (D.M.); goncalofloresft@outlook.com (G.F.); 2Research Center in Sport, Health and Human Development (CIDESD), 5001-801 Vila Real, Portugal; 3Research Unit for Sport and Physical Activity (CIDAF), Faculty of Sport Sciences and Physical Education (FCDEF), University of Coimbra, 3040-248 Coimbra, Portugal; geral.fernandasilva@gmail.com; 4Sport Sciences School of Rio Maior, Polytechnic of Santarém (ESDRM-IPSantarém), 2001-904 Rio Maior, Portugal; 5Department of Sports and Well-Being, Polytechnic Institute of Castelo Branco, 6000-266 Castelo Branco, Portugal; 6Sport Physical Activity and Health Research & Innovation Center, SPRINT, 2040-413 Santarém, Portugal

**Keywords:** quality of life, sedentary time, accelerometry, cardiovascular risk factors, sitting time, adults

## Abstract

Background: This systematic review analysed the association between objectively measured physical activity and sedentary behaviour with cardiovascular risk and HRQoL in adults without previous CVD. Additionally, we analysed the impact of the intensity of the physical activity in this association. Methods: The search was carried out in three electronic databases with access until February 2023 to find studies with an observational design. For quality assessment, we used The National Institute of Health (NIH) Quality Assessment Tool for Observational Cohort and Cross-Sectional Studies. Results: We identified 5819 references, but only five studies were included. One study shows a positive association between physical activity and HRQoL, while sedentary behaviour was negatively related to HRQoL. Another study showed an association between high-intensity physical activity with a better physical component of HRQoL and low-intensity physical activity with a better mental component of HRQoL. Three studies concluded that higher levels of physical activity are associated with lower levels of cardiovascular risk and higher levels of sedentary behaviour are associated with higher levels of cardiovascular risk. Conclusion: Our findings suggested that people who spend more time being active and spend less time being sedentary appear to have lower cardiovascular risk and higher HRQoL.

## 1. Introduction

Cardiovascular diseases (CVDs)—including coronary heart disease, acute myocardial infarction and stroke—are the most fatal non-communicable diseases globally [1,2]. CVD are the cause of 45% and 39% of all deaths in females and males, respectively, according to the most recent year of available data from the European Society of Cardiology member countries [3]. In order to prevent these diseases and minimise the related mortality, it is necessary to identify their risk factors and promote their reduction in the population. The most well-known risk factors are hypercholesterolemia, hypertension, smoking habits, diabetes mellitus, high body mass index, physical inactivity and sedentary behaviour [2,3,4,5,6]. To objectively evaluate cardiovascular risk, there are some models/algorithms validated that calculate the risk that each individual would have a cardiovascular event/disease in 10 years. Examples of this are the SCORE2 [1], the Framingham Risk Model [7] and the PROCAM [8]. In these three models/algorithms, the risk is calculated based on risk factors like sex, age, family history of CVD, cigarette smoking, systolic blood pressure, fasting HDL cholesterol, LDL cholesterol, total cholesterol, triglycerides and the fasting glucose concentration and diabetes mellitus diagnostic. The result is presented in a score or a percentage. 

Health-related quality of life (HRQoL) can be defined in several ways, as it is a multidimensional measure of physical, functional, mental and social wellbeing [9] that can be similar to quality of life or health status [10]. According to the available information, HRQoL is the value assigned to duration of life as modified by impairments, functional states, perceptions and social opportunities influenced by disease, injury, treatment or policy [11]. HRQoL can be assessed through diverse questionnaires like Short-Form Health Survey 36 (SF-36) [12] or WHOQOL-BREF [13].

Some of the risk factors mentioned above (i.e., hypertension, hypercholesterolemia, diabetes mellitus, high body mass index and sedentary behaviours) are also associated with low levels of HRQoL [14,15,16]. On the other hand, ideal cardiovascular health from early adulthood to middle age is associated with a higher HRQoL [17]. Even in adults with no history of CVD, lower levels of HRQoL are associated with a higher risk of incident CVD [18].

Increasing physical activity and reducing sedentary behaviour is highly recommended for all, though especially for people with high cardiovascular risk, since physical inactivity (when an individual fails to meet the World Health Organization’s physical activity recommendation [19,20]) and sedentary behaviours (SB) (i.e., any activity, in a position lying down, reclining or sitting, with an energy expenditure ≤ 1.5 metabolic equivalents (METs) [21,22]) are associated with a higher death rate [23,24]. Regarding the mechanisms, it is known that the absence of regular physical activity and the increased SB promotes visceral fat accumulation and, hence, an increased metabolic disorder, including metabolic syndrome [25,26,27]. Consequently, this represents an increase in risk factors, such as abdominal obesity, hyperglycaemia, hypertriglyceridemia, low high-density lipoprotein cholesterol and insulin resistance [28], contributing to the development of type 2 diabetes mellitus and CVD [5,6]. At the same time, older men who follow physical activity recommendations have high levels of HRQoL in the physical and mental domains [29]. On the other hand, individuals who spend more daily hours in SB have lower levels of HRQoL [16]. For people with cardiovascular risk, this association is also valid [30,31]. 

The association between physical activity with cardiovascular risk and HRQoL has been studied by some authors [16,29,31,32,33] and has been the subject of systematic reviews [30,34]. The authors found a beneficial association between physical activity and cardiovascular risk, as well as a beneficial association between physical activity and HRQoL, but had some limitations. Oguma and Shinoda-Tagawa [34] developed a systematic review and meta-analysis of the association between physical activity and cardiovascular risk. Still, in the inclusion criteria of the studies, they do not specify the physical activity assessment instrument, and they only include studies that evaluated the CVD-related morbidity and/or mortality, not the cardiovascular risk. Pucci et al. [30] made a systematic review of the association between physical activity and quality of life, but in the inclusion criteria of the studies, they do not specify the physical activity assessment instrument, and they include studies with different population groups (health and with diverse clinical conditions).

Accordingly, we intend to fill this gap in the literature and develop a systematic review with the following aim: to analyse the association between objectively measured physical activity and sedentary behaviour with cardiovascular risk and HRQoL in adults without previous CVD. Additionally, we intend to examine the impact of the intensity of the physical activity in this association.

## 2. Materials and Methods

This systematic review followed the Preferred Reporting Items for Systematic Reviews and Meta-analysis (PRISMA) guidelines [35]. The protocol of this systematic review was registered in the PROSPERO International Prospective Register of Systematic Reviews with the registration number CRD42023401025.

### 2.1. Search Strategies

The search was carried out in the following databases: Web of Science, PubMed and SCOPUS, with access until February 2023. In the databases Web of Science and PubMed, the search was made using free text in “All Fields”, and in the database SCOPUS, the search was made using free text terms only in “Article Titles, Abstract, Keywords”. MeSH terms were used when available. The search included studies with an observational design and was limited to studies published in English, Spanish and Portuguese. The grey literature and reference lists of the selected articles were also assessed to identify eligible articles. The literature search was performed according to the PICO strategy [36] and included a combination of free text and the Key Medical Subject Heading (MeSH terms), when available, as follows: (“cardiovascular risk” OR SCORE2 OR “cardiovascular disease risk factor” OR “CVD risk” OR “cardiometabolic risk factor” OR “clustered cardiometabolic risk” OR “composite score” OR “health related quality of life” OR “quality of life” OR HRQoL) AND (“physical activity levels” OR “moderate physical activity” OR “moderate-to-vigorous physical activity” OR “vigorous physical activity” OR “light physical activity” OR MVPA OR “energy expenditure” OR “sedentary behaviour” OR “sedentary behaviour” OR “sedentary time” OR “sitting time”) AND (acceleromet* OR “objectively measured” OR objective* OR “activity monitor” OR “heart rate monitor” OR pedomet* OR “motion sensor” OR device* OR count* OR armband* OR ActiGraph OR activPAL OR SenseWear OR Actical OR Actiheart) AND (adult* OR “middle aged” OR man OR men OR women OR woman).

### 2.2. Eligibility Criteria and Selection of Studies

The studies included in this review respect the following criteria created with the PICO (Population: adults without a history of cardiovascular disease, Intervention: objectively measured physical activity and/or sedentary behaviour assessment and Outcome: cardiovascular risk and HRQoL): (a) studies that investigated the association between physical activity and/or sedentary behaviour and cardiovascular risk and/or HRQoL in adults; (b) studies that evaluated cardiovascular risk objectively with SCORE2 or other validated scales/algorithms [1] that present the result in a score/percentage; (c) studies that evaluated HRQoL with WHOQOL-BREF [13] or other validated questionnaires; (d) studies that evaluated physical activity and sedentary behaviours objectively; (e) studies that include individuals without previous CVD; (f) studies in which the minimum age of the participants exceeds 18 years and the maximum age of participants does not exceed 65 years; (g) epidemiological studies of cross-sectional, observational, cohort and population-based designs and (h) peer-reviewed studies published in English, Spanish or Portuguese.

### 2.3. Data Extraction

Data from the search were imported into Endnote X9 (Thomson Reuters EndNote X9), and all duplicates were removed. The process of selection was performed in phases by two independent reviewers. Initially, the studies were selected based on the titles and abstracts. After this, the studies selected were reviewed in their entirety, taking into account the specific eligibility criteria. In case of a disagreement over the inclusion of articles, these were resolved through mediation by a third reviewer.

The extraction of data from the selected studies, as well as the extraction of the characteristics of these studies (name of the author, year of publication, the country in which it was carried out, the methodological design, characteristics of the sample, main outcomes and the instruments that are used to obtain the main outcomes) was performed independently by the two reviewers involved in the selection of the studies.

### 2.4. Quality and Risk of Bias Assessment

To evaluate the quality and risk of bias of the studies, The National Institute of Health (NIH) Quality Assessment Tool for Observational Cohort and Cross-Sectional Studies was used [37]. This checklist includes 14 items, and each one was classified as “yes” (Y), “no” (N), “not applicable” (NA), “not reported” (NR) or “cannot determine” (CD). Based on the number of items classified with “yes”, the total of items applicable for each study was calculated by a total score and a percentage. Based on this percentage, the studies were classified according to quality rating: poor < 50%, fair 50–75% and good > 75% [38].

This assessment would not be intended as a condition for the study to be included in this systematic review but rather to identify studies in which a poor-quality assessment could interfere with the outcomes. The quality scoring was performed by two independent reviewers, and in case of a disagreement, these were resolved by a third reviewer.

## 3. Results

### 3.1. Data Search

The sequence followed for the selection of the studies that were included in this systematic review is shown in Figure 1. 

The initial search in the database identified a total of 5819 potentially eligible studies. After excluding studies based on duplicates (n = 2234), titles and abstracts (n = 3566), 19 full-text articles were analysed according to the inclusion and exclusion criteria. Most of the studies excluded were because the maximum age of the participants was over 65 or was not described. After this phase, 14 studies were excluded, and 5 were included in this systematic review.

### 3.2. Characteristics of Studies

Details of the five studies included in the systematic review are presented in Table 1.

Four of the studies were conducted in European countries from Southern Europe; Western Europe and North Europe (Spain, United Kingdom, Sweden and Finland), and one was conducted in Australia. One of the studies included only female participants [39], while the other four studies included participants from both sexes [40,41,42,43], but only one of them analysed the differences between genders. The studies included a total sample of 6191 participants. The ages of all participants varied between 18 and 65 years, but each study had different age intervals. All studies included had a cross-sectional design.

All the studies evaluated the physical activity and sedentary behaviour objectively with different devices (ActivPAL monitor; Polar Active, Polar Electro Oy, Kempele, Finland; ActiGraph GT3X and GT3X+, ActiGraph, LCC, Pensacola, FL, USA). Tigbe et al. [40] analysed the time that the participants spent stepping, standing and sitting/lying, as well as steps, mean stepping rate and number of sit-to-stand transitions per day, while the other authors [39,41,42,43] analysed the sedentary time (ST); time spent in different intensity’s of physical activity (light, moderate, moderate-vigorous (MVPA) and vigorous physical); bout of MVPA and bout of ST. However, each study has its definition of bouts and cut points. The definition of the different levels of physical activity intensity is different in each study, because the cut points are distinct. However, according to Ainsworth et al. [44], examples of low-intensity activities are walking slowly and preparing food; on the other hand, examples of vigorous activities are running and carrying heavy loads. HRQoL was evaluated in two of the included studies [39,43]. Marín-Jiménez et al. [39] used the Short-Form Health Survey 36 (SF-36), a questionnaire with 36 items grouped into eight dimensions: physical functioning, role limitations due to physical health (role-physical), bodily pain, general health perceptions, vitality, social functioning, role limitations due to emotional problems (role-emotional) and mental health. Examples of role-physical items include “cut down on the amount of time you spent on work or other activities” or “accomplished less than you would like”, and examples of role-emotional items include “cut down the amount of time you spent on work or other activities” or “didn’t do work or other activities as carefully as usual”. The scores range from 0 to 100 in every dimension, where higher scores indicate better health. These eight dimensions can be summarised into two global concepts: the physical component scale (covered by physical functioning, role-physical, bodily pain and general health perceptions) and the mental component scale (vitality, social functioning, role-emotional and mental health) [12,45]. Kolt et al. [43] used the five-item “general health” subscale of the RAND 36-Item Health Survey (RAND-36), a license-free instrument developed from the original SF-36 Medical Outcomes Study survey. All items are scored on a scale of 0 to 100, with a higher score indicating a more favourable health state [46]. Kobayashi Frisk et al. [42], Niemelä et al. [41] and Tigbe et al. [40] analysed the association of physical activity with cardiovascular risk and used three different instruments to evaluate the cardiovascular risk. Kobayashi Frisk et al. [42] used SCORE2, a European risk scoring model that estimates the 10-year risk of first-onset CVD based on gender, age, smoking status, systolic blood pressure and non-high-density lipoprotein cholesterol. Being a prediction algorithm that considers both fatal and non-fatal CVD, SCORE2’s results are reported as a percentage. Niemelä et al. [41] used the Framingham risk model, which estimates the absolute risk over 10 years of overall CVD. The variables used in this model include age, HDL cholesterol, total cholesterol, systolic blood pressure (not treated or treated) and prevalence of smoking (yes/no) and diabetes mellitus (yes/no). Being an estimate tool, the result is presented by a percentage. Tigbe et al. [40] used PROCAM, a risk calculator that generates 10-year coronary heart disease risk for men aged 35–65 y.o. and women aged 45–65 y.o., based on sex, age, family history of coronary heart disease, cigarette smoking, systolic blood pressure, fasting HDL cholesterol, LDL cholesterol, triglycerides and fasting glucose concentration. Additionally, participants were classified as having metabolic syndrome or not using the following criteria: fasting serum triglycerides ≥ 1.7 mmol/L, glucose ≥ 5.6 mmol/L, HDL cholesterol ≤ 1.03 mmol/L for men or ≤1.30 mmol/L for women, waist circumference ≥ 102 cm for men or ≥88 cm for women and blood pressure ≥ 130/85 mmHg. In this study, only 67 men and 6 women have the adequate age to use PROCAM, and the other participants have the presence or absence of metabolic syndrome as the indicator of cardiovascular risk.

Marín-Jiménez et al. [39] concluded that lower ST and greater light physical activity were associated with a better SF-36 emotional role. Higher MVPA was associated with a better SF-36 physical function and SF-36 vitality. Higher vigorous physical activity was associated with a better SF-36 physical function, SF-36 bodily pain and the SF-36 physical component scale. Finally, moderate physical activity was not associated with any SF-36 dimension. In the study of Kolt et al. [43], the association of physical activity and sedentary behaviour with HRQoL was analysed, taking into account the duration and frequency of the physical activity. The authors concluded the duration measure (average daily minutes) of physical activity was positively related to general HRQoL. In contrast, the physical activity bouts (consecutive 10-min period) measure was negatively associated with general HRQoL. At the same time, the duration measure (average daily minutes) of sedentary behaviour was negatively related to general HRQOL. The frequency measure of sedentary behaviour was not significantly associated with general HRQoL.

Kobayashi Frisk et al. [42] divided the participants into groups based on their chronotype (extreme morning, moderate morning, intermediate, moderate evening and extreme evening). The groups with higher cardiovascular risk are those with longer spent sedentary, with a lower average daily physical activity, lower percentage of light-intensity physical activity and a lower percentage of MVPA. The group with the best results is the chronotype “extreme morning”, and the group with the worst results are the chronotype “extreme evening”. Niemelä et al. [41] divided the participants into four clusters (inactive, evening active, moderately active and very active) based on the pattern of activity. The ST is higher in the “inactive” cluster and lower in the “very active” cluster. On the other hand, all the variables related to physical activity are higher in the “very active” cluster and lower in the “inactive” cluster. Between the clusters “evening active” and “moderately active”, the second one had healthy results, slightly below the “very active” cluster results. Regarding the cardiovascular risk, the men in the “inactive” cluster have higher cardiovascular risk results, and the men in the “very active” cluster have a lower cardiovascular risk; the women in the “evening active” (3.75%) and “inactive” (3.0%) clusters have a higher cardiovascular risk, and the women in the “moderately active” (2.99%) and “very active” (3.06%) clusters have a lower cardiovascular risk [41]. Tigbe et al. [40] concluded that higher cardiovascular risk is significantly associated with ST (in sitting/lying positions), and on the other hand, cardiovascular risk is significantly and favourably associated with stepping time; additional cardiovascular risk is inversely associated with the daily steps count.

According to the results of Kobayashi Frisk et al. [42] and Niemelä et al. [41], the clusters with lower cardiovascular risk are the clusters that presented a higher percentage/time per day of physical activity at different levels of intensity (light physical activity, MVPA, vigorous physical activity and very vigorous physical activity) simultaneously, but, in all the clusters, the time spent in light physical activity is much higher than in MVPA, suggesting that people who spend more time being active have a lower cardiovascular risk, even if it is low-intensity physical activity. The statistical results of each included study are presented in Table 1.

**Table 1 healthcare-12-01866-t001:** Characteristics of the 5 included studies.

Author, Year, Country	Sample Size (n Total; n ♂/n ♀)	Age (Years) (Mean ± SD; Range)	Study Design	Sedentary Behaviour/Physical Activity Assessment	Health Related Quality of Life (HRQOL) Assessment	Cardiovascular Risk Assessment	Main Outcomes	Main Goals	Main Results	Quality and Risk of Bias Assessment
1. Marín-Jiménez et al. [1] Spain Fitness League Against MENopause COst (FLAMENCO) project	182 (182 ♀)	52.6 ± 4.5 (45–60 y)	Cross-sectional study	Device: GT3X, Pensacola, FL; Days of wear: 9 days, but the first and the last was excluded from the analyses Minimum wear: Not applicable (N/A) Epochs: N/A Cut points: N/A Parameters evaluated: Sedentary time (ST), time in light, moderate, moderate-vigorous (MVPA), and vigorous physical activity (PA), total PA time per day and per week, bouted MVPA (period of 10 or more consecutive minutes (min) of duration in MVPA) and percentage of participants who met the international PA recommendations of at least 150 min of MVPA per week	Short-Form Health Survey 36 (SF-36) (score)	_________	Weight, Height, body mass index (BMI), ST, PA and health-related quality of life (HRQoL)	To analyse the association of ST and PA with HRQoL in middle-aged women	Lower ST and greater light PA were associated with a better SF-36 emotional role (B: −0.03; 95% confidence interval (CI): −0.07 to −0.00; *p* = 0.02 and B: 0.04, 95% CI: 0.00–0.08; *p* = 0.01, respectively). Higher MVPA was associated with a better SF-36 physical function (B: 0.01, 95% CI: 0.00–0.02; *p* = 0.05) and SF-36 vitality (B: 0.02, 95% CI: 0.00–0.03; *p* = 0.01). Higher vigorous PA was associated with a better SF-36 physical function (B: 0.34, 95% CI: 0.0–0.66; *p* = 0.03), SF-36 bodily pain (B: 0.63, 95% CI: 0.02–1.25; *p* = 0.04), and the SF-36 physical component scale (B: 0.20, 95% CI: 0.00–0.39 *p* = 0.04). Higher total PA was associated with a better SF-36 emotional role (B: 0.03, 95% CI: 0.00–0.07: *p* = 0.02).	9/12 (75%)
2. Tigbe et al. [2] United Kingdom	111 (96 ♂/15 ♀)	39 ± 8 ♂/42 ± 9 ♀ (22 to 60 y)	Cross-sectional study	Device: ActivPAL monitor; Days of wear: 7 consecutive days; Minimum wear: three 24-h periods, including a non-work day Epochs: N/A Cut points: N/A Parameters evaluated: time spent stepping, standing and sitting/lying as well as steps, mean stepping rate and number of sit-to-stand transitions per day.	________	PROCAM (score) Presence of the metabolic syndrome using the following specific criteria	PA, weight, height, waist circumference and CHD risk	To examined the associations between CHD risk and time spent in objectively- measured postures (sitting, lying and standing) and of stepping	Higher 10-year PROCAM risk was significantly (*p* < 0.05) associated with ST adjusting for age, sex, Scottish Index of Multiple Deprivation (SIMD), family history of CHD, job type and shift worked.	7/12 (58.3%)
3. Niemelä et al. [3] Finland	4582 (1916 ♂/2666 ♀)	(46–48 y)	Cross-sectional	Device: Polar Active, Polar Electro Oy, Kempele Finland; Days of wear: 14 days Minimum wear: 7 consecutive days with enough PA data (wear time ≥ 600 min/day), starting from the second measured day; Epochs: N/A Cut points: very light: 1–1.99 MET, light: 2–3.49 MET, moderate: 3.5–4.99 MET, vigorous: 5–7.99 MET, and vigorous+ ≥8 MET; MVPA was assessed as all activity at least 3.5 METs, while ST was assessed as the duration of very light activity Parameters evaluated: Daily averages of time spent in different activity levels; Total daily duration obtained in MVPA and ST bouts (at least 30 min of consecutive MET values between 1 and 2 METs).	________	Framingham risk model (percentage)	Height, weight, BMI, body fat percentage and visceral fat area, total cholesterol, low-density lipoprotein (LDL) and high-density lipoprotein (HDL) cholesterol levels, Systolic (SBP) and diastolic blood pressures (DBP), PA, CVD risk,	To identify temporal patterns of continuously measured physical activity beneficial for cardiovascular health in a middle-aged group using cluster analysis and to study how the widely used 10-year CVD risk model is associated with different PA profiles.	Significant differences in CVD risk between clusters were found both in men (*p* = 0.028) and women (*p* < 0.001). The inactive cluster had higher CVD risk compared with the very active cluster in men (*p* < 0.05). In women, the inactive cluster had higher CVD risk compared to moderately active and very active clusters, and the evening active cluster had higher risk compared to the moderately active cluster (*p* < 0.05).	8/12 (66.7%)
4. Kobayashi Frisk et al. [4] Sweden	812 (48% ♂/52% ♀)	57.6 ± 4.4 (50–64 y)	Cross-sectional analysis	Device: ActiGraph GT3X and GT3X +, ActiGraph, LCC, Pensacola, FL, USA. Days of wear: 7 consecutive days Minimum wear: at least 600 min per day of wear time for at least 4 days Epochs: N/A Cut points: time spent sedentary (SED): 0–199 cpm, time spent in light intensity physical activity (LIPA): > 199 & < 2690 cpm, and time spend in moderate to vigorous intensity physical activity (MVPA): ≥ 2690 cpm Parameters evaluated: Daily percentage of SED and MVPA, total volume of physical activity (mean cpm of wear time), bout of SED (at least 20 min of consecutive cpm values <199 with no allowance for interruption above the threshold), bout of MVPA (10 min consecutive ≥ 2690 cpm, with an allowance of up to 2 min below this threshold), percentages of SED and MVPA in the morning (06:00 to 12:00), afternoon (12:00 to 18:00) and evening (18:00 to 00:00)	________	SCORE2 (score)	Chronotype, Mid-sleep time, Subjective sleep quality, Habitual sleep duration, PA, SED, Estimation of the 10-year risk of frst-onset CVD,	To investigate the relationship between chronotype, objectively measured physical activity patterns, and 10-year frst-onset CVD risk assessed by the Systematic Coronary Risk Evaluation 2 (SCORE2)	Extreme evening chronotypes exhibited the most sedentary lifestyle and least MVPA (55.3 ± 10.2 and 5.3 ± 2.9% of wear-time, respectively). Extreme evening chronotype was associated with increased SCORE2 risk compared to extreme morning type independent of confounders (β = 0.45, SE = 0.21, *p* = 0.031). SED was a significant mediator of the relationship between chronotype and SCORE2.	8/12 (66.7%)
5. Kolt et al. [5] Australia WALK 2.0 randomised controlled trial	504 (176 ♂/328 ♀)	50.8 ±13.1 (18–65 y)	Cross-sectional	Device: ActiGraph GT3X activity monitor Days of wear: 7 consecutive days Minimum wear: 10 h of wear time on at least 5 days in the 7 day period. Epochs- 1 s Cut-points: MVPA—more than 1951 counts/min; Sedentary behaviour—less than 100 counts/min; Parameters evaluated: Daily measures of MVPA, sedentary behaviour, bouts (consecutive 10-min period) of MVPA, bouts of sedentary time and wear time.	5-item ‘general health’ subscale of the RAND 36-Item Health Survey (RAND-36) (Score)	_________	PA, Sedentary behaviour (SB), HRQoL	To examine the association of HRQoL with PA and sedentary behaviour, using both continuous duration (average daily minutes) and frequency measures (average daily number of bouts ≥10 min).	The duration measure (average daily minutes) of physical activity was positively related to general HRQoL (path coefficient = 0.294, *p* < 0.05) after adjusting for covariates of age, gender, BMI, level of education, and activity monitor wear time. In contrast, the physical activity bouts measure was negatively related to general HRQoL (path coefficient = −0.226, *p* < 0.05) after adjusting for covariates. The duration measure (average daily minutes) of sedentary behaviour was negatively related to general HRQoL (path coefficient = −0.217, *p* < 0.05) after adjusting for covariates of age, gender, BMI, level of education, and activity monitor wear time.	8/12 (66.7%)

### 3.3. Quality and Risk of Bias Assessment

The quality and risk of bias of the studies were evaluated, and the results are shown in Table 1.

All the studies are cross-sectional, and because of this, items 6 and 7 were classified with “No” according to the guidance [37]. Items 12 and 13 were classified as “Not applicable” and were not accounted for in the final score. The final percentage varied between 58.3% and 75%, corresponding to the quality rating “fair”. 

## 4. Discussion

This systematic review analysed the association between objectively measured physical activity and sedentary behaviour with cardiovascular risk and HRQoL in adults without previous CVD. At the same time, we intended to analyse the impact of the intensity of the physical activity in this association. 

As expected, all the studies show a positive association between physical activity and HRQoL [39,43] and between physical activity and cardiovascular risk [40,41,42]. On the other hand, all the studies show a negative association between sedentary behaviour with HRQoL [39,43] and between sedentary behaviour and cardiovascular risk [40,41,42]. These results generally support the concurrent validity of the questionnaires used in the studies. Also, evidence of predictive validity was found in several studies.

About the impact of physical activity intensity on the HRQoL, the results of Marín-Jiménez et al. [39] indicated that higher intensity is associated with the physical component of HRQoL and lower intensity is associated with the mental component of HRQoL (which refers to aspects of an individual’s psychological well-being and mental health). These results are consistent with the results of Vásquez et al. [47], a study that analysed the association between MVPA with self-reported mental HRQoL and physical HRQoL in adults and older ages. Vásquez et al. [47] observed no significant linear trend between accelerometer-measured MVPA and mental HRQoL and a significant positive association between MVPA and physical HRQoL, where a higher MVPA corresponded with higher scores in physical HRQoL. The results of Kolt et al. [43] suggested that a longer duration of physical activity and sedentary behaviour are associated with higher and lower HRQoL, respectively. Regarding frequency, the results indicate that, for a given level of physical activity duration, being active in fewer bouts was associated with a better HRQoL [43]. These results are consistent with a previous study that demonstrated that both non-bouted and bouted MVPA are associated with HRQoL. However, the authors had already found a low level of engagement in bouted MVPA [48]. Guallar-Castillón et al. [49] also studied the association of physical activity and sedentary behaviour with HRQoL in adults and older ages, using a pattern of activity, and his results are consistent with the results of the included articles of this systematic review [39,43]. Guallar-Castillón et al. [49] concluded that a pattern with vigorous physical activity was associated with better physical health, and the pattern with light physical activity was associated with better mental health. Additionally, the patterns that included some types of physical activity (vigorous or light) were associated with a better HRQL compared to the pattern that included more sedentary behaviour. 

In a general way, it was expected that physical activity, independently of its duration, intensity or frequency, was associated with a better HRQoL, because physical activity is associated with many health-related outcomes like low mortality [50], low risk of depression and anxiety [51], improvements in cognition [52] and better sleep quality [53] and with a low risk of many diseases, like cancer, hypertension and type 2 diabetes [19].

In the studies of Kobayashi Frisk et al. [42] and Niemelä et al. [41], the participants were in groups/clusters. Both concluded that the participants with more average daily physical activity, with less ST, with more percentage of light-intensity physical activity and with more average time in MVPA are the group of participants with lower cardiovascular risk. These results are consistent with other studies that have already analysed the association between physical activity and cardiovascular risk and concluded that higher levels of physical activity are associated with lower cardiovascular risk [54,55]. Tigbe et al. [40] concluded that higher cardiovascular risk is significantly associated with ST (in sitting/lying positions), and on the other hand, cardiovascular risk is significantly and favourably associated with stepping time; additional cardiovascular risk is inversely associated with the daily steps count. These results are similar to the other authors who analysed the daily step counts with cardiovascular risk or cardiovascular mortality and concluded that an increase in the number of daily steps represents a decrease in cardiovascular risk or cardiovascular mortality [56,57].

As is already known, physical activity is very important in the control of risk factors for numerous diseases, and CVD is not an exception. Physical activity is recommended by the World Health Organization [20] and by the European Society of Cardiology [2] in order to reduce cardiovascular risk and in order to control the risk factors for CVD like diabetes mellitus [57], high body mass index, hypertension and hypercholesterolemia [58]. Following this, it is expected that more active people will have less risk factors and lower cardiovascular risk.

The results of the included studies reinforce the importance of increasing the levels of physical activity and reduced the levels of ST in the population to reduce cardiovascular risk and, consequently, the death rate for CVD, as well as to promote the increase of the levels of HRQoL in the population. 

This systematic review has great methodological value due to the eligibility criteria guaranteeing that only studies involving adult participants and using objective measures of physical activity and sedentary behaviour were included. Also, only studies using validated instruments to measure the cardiovascular risk and HRQoL were eligible for the study. However, some limitations might be considered. The limitations of this study are the fact that the research was carried out using terms only in English; the limited number of studies that used objective instruments to measure physical activity and validated instruments to measure cardiovascular risk; the limited number of studies that included only adults and not adults and older people; the heterogenicity of the instruments that evaluated the cardiovascular risk and the HRQoL; the heterogenicity of the protocols to collect, process and analyse the data from accelerometers and the use of different device to evaluate physical activity.

In the future, it will be interesting to study the mediator role of physical activity in the relationship between HRQoL and cardiovascular risk, as well as the association of each component of physical activity (frequency, intensity, duration and type) with cardiovascular risk and HRQoL, to know what represents more benefits to the population and to guide the physical activity recommendations and the exercise prescriptions.

## 5. Conclusions

This systematic review contributed to the continuation of the discussion on increasing levels of physical activity and reducing levels of ST in the population, since our findings suggested that people who spend more time being active and spend less time being sedentary apparently have lower cardiovascular risk and higher HRQoL. Considering the physical activity intensity, higher-intensity physical activity appears to be associated with a higher physical component of HRQoL, and lower-intensity physical activity appears to be associated with a higher mental component of HRQoL. Cardiovascular risk tends to be lower in more active people, even if it is low-intensity physical activity.

Despite the limitations of this systematic review and the need to continue studying these associations, the results reinforce the importance of sharing the benefits of physical activity with the population and the importance of promoting a behavioural change in the population.

## Figures and Tables

**Figure 1 healthcare-12-01866-f001:**
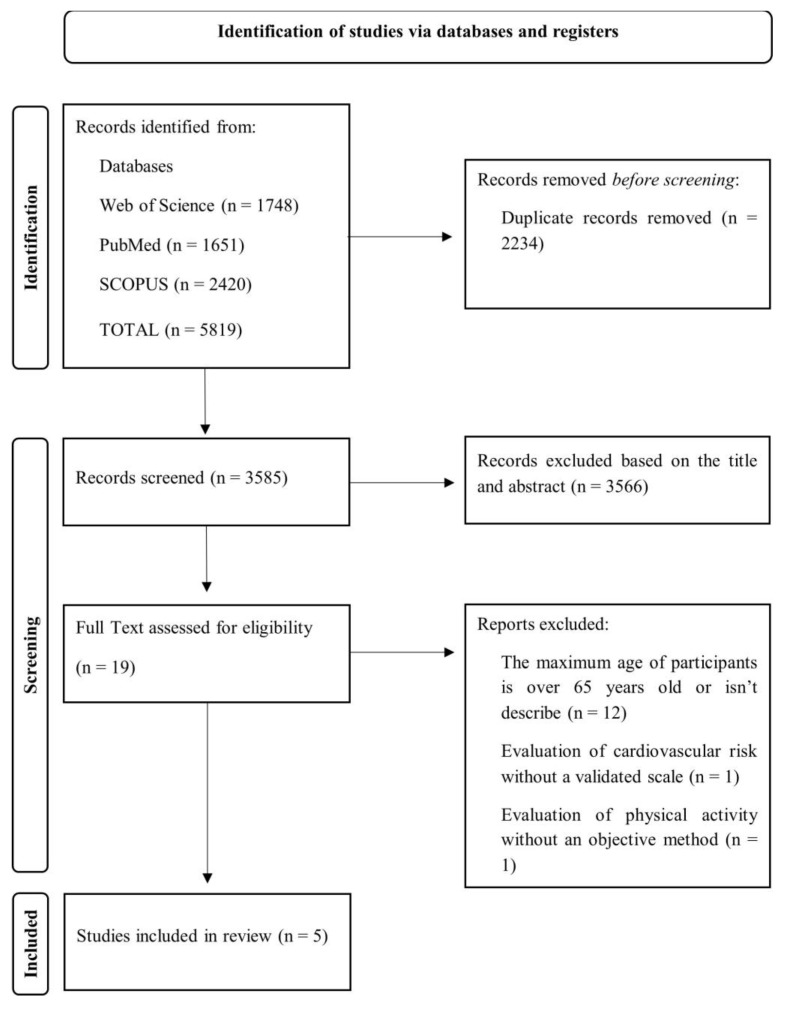
Search strategy and studies selection flow chart.

## Data Availability

No new data were created or analysed in this study. Data sharing is not applicable to this article.

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
