# Peer review of "Objectively Measured Physical Activity and Sedentary Behaviour on Cardiovascular Risk and Health-Related Quality of Life in Adults: A Systematic Review"

_healthcare, 2024, doi:10.3390/healthcare12181866_

Round 1

Reviewer 1 Report

Comments and Suggestions for Authors

Comments to both author and editor

The authors clearly demonstrate their knowledge and experience with the impact of sedentary behavior on the actual risk of pathophysiological processes and outcome, e.g. systolic blood pressure, fasting HDL-cholesterol, LDL-cholesterol, total cholesterol, triglycerides and fasting glucose concentration, diabetes mellitus, etc.

While there are established advantages to using their criteria-specific small sample in this almost meta-analysis,  it is obviously more difficult to obtain significant statistical differences  if and when necessary with only five studies.  However, the elimination of many studies in accordance with inclusion criteria minimized the presence of possible extraneous variables which would likely obscure important results.

Unfortunately, however, the English language in the manuscript needs help.  Word choice and phrasing are often not correct.   Some specific language problems include:

                The word “simultaneous” below is confusing.

“ According to the results of Kobayashi Frisk et al.[42]  and  Niemelä et al.[41], the clusters with lower cardiovascular risk are the clusters that presented a higher percentage/ time per day of light physical activity, MVPA, vigorous physical activity and very vigorous physical activity, simultaneous.”

and the following is not a sentence and thus not clear:

“What is consistent with the results of one study that analyses the association between MVPA with self-reported mental HRQoL, and physical HRQoL in adults and older age [46].”

In addition to the manuscript needing a table of acronyms, the second sentence below does not appear to follow from the first sentence

“The ST is higher in “inactive “cluster and lower in the “very active “cluster. In other hand, all the variables related to the physical activity are higher in the “very 331 active” cluster and lower in the “inactive” cluster.”

Given the possibility of gender, age, ethnocultural, and geographic differences, the authors might consider providing better organized demographic information.

Further explanation or examples would also be helpful of “low-intensity exercise”; “SF 36 emotional role”, and of “mental component of HRQoL” .

Perhaps it would be interesting to mention the validity and reliability of self-reported behaviors in this report.

Together with the results of this extensive search of the literature prior to reporting the in-depth analysis of the sedentary risk factor, this manuscript clearly would be better as a chapter in a medical textbook emphasizing the importance of physical activity to lower CVD risk.

Update:

See my responses in italics below your questions

• What is the main question addressed by the research? 

The main question addressed the association of physical activity and sedentary behavior with cardiovascular risk.

• Do you consider the topic original or relevant to the field? Does it address a specific gap in the field? Please also explain why this is/ is not the case.

It is not original but it is relevant because of the inclusion/ exclusion criteria; See SECTION 2.2 .  For example, it included only “studies in which the minimum age of participants exceeds 18 years and the maximum age of participants does not exceed 65 years”

• What does it add to the subject area compared with other published material?

It adds the inclusion/ exclusion areas noted above.

• What specific improvements should the authors consider regarding the methodology? What further controls should be considered?

              It was hard at first to understand the justification for analyzing only five studies out of over 5,000 because the manuscript initially seemed to be written more in the format of presenting five cases.  Although the authors are to be complimented on their thoroughness in looking at the subject, it cannot be called a classic research design.  However, it is all right for a systematic review.

              In fact, it is a superb presentation of scientifically and clinically useful material.  The format of the study just did not seem appropriate to this reviewer. On the other hand, it may be the only way of communicating the results of their search and could be identified as such, so my comment that  the research design “must be improved” was based on expectations for a meta-analysis which this manuscript is not.  It differs from a meta-analysis which would analyze almost everything that is out there on a particular subject to give the reader an idea of the scope of the problem, whereas this manuscript does a great service of making the best representation of the relationship between sedentary behavior, physical activity, and cardiovascular risk.

              Therefore, I withdraw my comment “must be improved” with the suggestion that perhaps a few words about the presentation in this useful but not traditional format would be in order.   Unless the editor feels there is any merit to my suggestion of format, I withdraw my concerns about the content of the manuscript but still maintain there might be a more appropriate format and that it needs appropriate editorial textual and other revisions.

• Are the conclusions consistent with the evidence and arguments presented and do they address the main question posed? Please also explain why this is/is not the case.   

My answer to the multiple choice question was yes, “the conclusions are supported by the results”

• Are the references appropriate?  

Yes, very much so.

• Any additional comments on the tables and figures. 

None.

Comments on the Quality of English Language

See above.

Author Response

RESPONSE TO REVIEWER 1

The authors clearly demonstrate their knowledge and experience with the impact of sedentary behavior on the actual risk of pathophysiological processes and outcome, e.g. systolic blood pressure, fasting HDL-cholesterol, LDL-cholesterol, total cholesterol, triglycerides and fasting glucose concentration, diabetes mellitus, etc.

While there are established advantages to using their criteria-specific small sample in this almost meta-analysis, it is obviously more difficult to obtain significant statistical differences if and when necessary, with only five studies.  However, the elimination of many studies in accordance with inclusion criteria minimized the presence of possible extraneous variables which would likely obscure important results.

Authors' response: Thank you for the valuable comments. In fact, the inclusion criteria were essential to eliminate studies that could distort the results but with only five studies which analyze the variables in different ways was not possible proceed to a meta- analysis.

Unfortunately, however, the English language in the manuscript needs help.  Word choice and phrasing are often not correct.   Some specific language problems include:

                The word “simultaneous” below is confusing.

“ According to the results of Kobayashi Frisk et al.[42]  and  Niemelä et al.[41], the clusters with lower cardiovascular risk are the clusters that presented a higher percentage/ time per day of light physical activity, MVPA, vigorous physical activity and very vigorous physical activity, simultaneous.”

and the following is not a sentence and thus not clear:

“What is consistent with the results of one study that analyses the association between MVPA with self-reported mental HRQoL, and physical HRQoL in adults and older age [46].”

 Authors' response: We are thankful for the reviewer’s comment. We have restructured the two sentences.

Action: The first sentence was restructured and the word “simultaneous” was removed. The second sentence was restricted as well in order to clarify the information and the connection with the previous sentence and with the following sentence.

In addition to the manuscript needing a table of acronyms, the second sentence below does not appear to follow from the first sentence

“The ST is higher in “inactive “cluster and lower in the “very active “cluster. In other hand, all the variables related to the physical activity are higher in the “very active” cluster and lower in the “inactive” cluster.”

Authors' response: We are thankful for the reviewer’s comment. We create a list of acronyms.

About the sentences, the intention is demonstrating the results of one study and explain what is more common in each cluster.   

Actions: We create a list of acronyms and are submit with the manuscript.

Given the possibility of gender, age, ethnocultural, and geographic differences, the authors might consider providing better organized demographic information.

Authors' response: We are thankful for the reviewer’s comment. We reorganize the first paragraph of the results.

Action: We add some information in order to clarify the demographic differences between the samples of the studies.

Further explanation or examples would also be helpful of “low-intensity exercise”; “SF 36 emotional role”, and of “mental component of HRQoL” .

Authors' response: We are thankful for the reviewer’s comment. We clarify the definitions and/ or include some examples in the manuscript.

Actions: Lines 190-193 include examples of low intensity activities; lines 196-198 terms were corrected to include more adequate terms and examples (“physical role” was altered to role limitations due to physical health (role-physical); body pain altered to bodily pain; general health was altered to general health perceptions; and emotional role altered to role limitations due to emotional problems (role-emotional); and lines 204-206 were also corrected accordingly; in lines 291-292, examples of mental component of HRQoL were included.

Perhaps it would be interesting to mention the validity and reliability of self-reported behaviors in this report.

Authors' response: Thank you for your comments. A mention to validity and reliability  was included in lines 289-290.  

Together with the results of this extensive search of the literature prior to reporting the in-depth analysis of the sedentary risk factor, this manuscript clearly would be better as a chapter in a medical textbook emphasizing the importance of physical activity to lower CVD risk.

Authors' response: We are thankful for the reviewer’s valuable comment.

Update:

See my responses in italics below your questions

  • What is the main question addressed by the research?

The main question addressed the association of physical activity and sedentary behavior with cardiovascular risk.

  • Do you consider the topic original or relevant to the field? Does it address a specific gap in the field? Please also explain why this is/ is not the case.

It is not original but it is relevant because of the inclusion/ exclusion criteria; See SECTION 2.2.  For example, it included only “studies in which the minimum age of participants exceeds 18 years and the maximum age of participants does not exceed 65 years”

  • What does it add to the subject area compared with other published material?

It adds the inclusion/ exclusion areas noted above.

  • What specific improvements should the authors consider regarding the methodology? What further controls should be considered?

              It was hard at first to understand the justification for analyzing only five studies out of over 5,000 because the manuscript initially seemed to be written more in the format of presenting five cases.  Although the authors are to be complimented on their thoroughness in looking at the subject, it cannot be called a classic research design.  However, it is all right for a systematic review.

              In fact, it is a superb presentation of scientifically and clinically useful material.  The format of the study just did not seem appropriate to this reviewer. On the other hand, it may be the only way of communicating the results of their search and could be identified as such, so my comment that the research design “must be improved” was based on expectations for a meta-analysis which this manuscript is not.  It differs from a meta-analysis which would analyze almost everything that is out there on a particular subject to give the reader an idea of the scope of the problem, whereas this manuscript does a great service of making the best representation of the relationship between sedentary behavior, physical activity, and cardiovascular risk.

              Therefore, I withdraw my comment “must be improved” with the suggestion that perhaps a few words about the presentation in this useful but not traditional format would be in order.   Unless the editor feels there is any merit to my suggestion of format, I withdraw my concerns about the content of the manuscript but still maintain there might be a more appropriate format and that it needs appropriate editorial textual and other revisions.

  • Are the conclusions consistent with the evidence and arguments presented and do they address the main question posed? Please also explain why this is/is not the case.

My answer to the multiple choice question was yes, “the conclusions are supported by the results”

  • Are the references appropriate?

Yes, very much so.

  • Any additional comments on the tables and figures.

None.

Authors' response: Thank you for the valuable comments and suggestions for improving our paper.

Reviewer 2 Report

Comments and Suggestions for Authors

My comments are attached. 

Comments on the Quality of English Language

The English needs to be revised. See the attached document for English mistakes. 

Author Response

RESPONSE TO REVIEWER 2

This study examined five previous studies that objectively measured physical activity and sedentary behavior in healthy adults. The results of this systematic review indicate a positive relationship with physical activity and health-related quality of life and an inverse relationship with sedentary behavior and health-related quality of life. Overall, this is a good and useful study design with a novel approach to previous data.

However, there are several grammar issues. I have listed some of the multiple grammar issues below along with other comments regarding the paper.

-Line 59: Change “In other hand” to “On the other hand”

-Line 62: Change “Increase physical activity and reduce sedentary behaviour” to “Increasing physical activity and reducing sedentary behaviour”

-Line 67: Remove the spacing before “Regarding”.

-Line 136: Add “the” between “from” and “search”

-Line138: Change “Initial” to “Initially”

-Lines 148-149: Move “was used” to the end of the sentence.

-Line 152: Remove “was”

-Line 167: Remove the spacing before “Most”

-Line 168: Change to “excluded were”

-Line 175: add “the” between “of” and “studies”

-Line 179: Decrease the space before “All”

-Line 185: “Analyse” should be “Analyzed”.

-Line 196: Delete the space before “used”

-Lines 212-218: Please discuss more information here about what was used to classify as having metabolic syndrome. For example, how many of these would classify one as having metabolic syndrome.

Authors' response: Explanation was included in line 215-216 and 220.

-Line 219: Delete the space before “conclude”

-Line 224: Delete the space before “the”

-Lines 240-241: Change “In other hand” to “On the other hand”

-Lines 245-247: Fix the grammar in these sentences.

-Line 252: Decrease the space before “and”

-Line 255: Should “simultaneous” be “simultaneously”?

-Lines 257: Change “been” to “being”

-Lines 269: Change “intend” to “intended”

-Lines 271-272: PA and CV risk should have a negative association.

Authors' response and action: We hope that clarifications made to the text reflect the correct associations we intended to report in the first place. 

-Lines 273-274: SB and CV risk should have positive association.

Authors' response and action: We hope that clarifications made to the text reflect the correct associations we intended to report in the first place. 

-The remainder of the discussion needs English grammar edits.

Authors' response: We are thankful for the reviewer’s comment and corrections for improving our paper.

Action: We correct all the suggestions on the manuscript.

Reviewer 3 Report

Comments and Suggestions for Authors

In this manuscript (healthcare-3164591), the authors analyzed the correlation between objectively measured physical activity and sedentary deeds with cardiovascular risk and health related quality of life (HRQoL) without preceding CVD. They suggested that people who spend more time being active and spend less time being inactive appear to have less cardiovascular risk and high HRQoL. 

Minor concerns

  1. In the study have authors noticed any relation between gender difference and health related quality of life?
  2. In line 85, please correct “maded” to “made”
  3. Please correct the Ref.51

Author Response

RESPONSE TO REVIEWER 3

In this manuscript (healthcare-3164591), the authors analyzed the correlation between objectively measured physical activity and sedentary deeds with cardiovascular risk and health related quality of life (HRQoL) without preceding CVD. They suggested that people who spend more time being active and spend less time being inactive appear to have less cardiovascular risk and high HRQoL.

Minor concerns

1.In the study have authors noticed any relation between gender difference and health related quality of life?

Authors' response: We understand that this would be very interesting to be included in the analysis. However, due to the limited number of studies, sample characteristics included in the studies and reported variables, we were not able to perform this analysis.

2.In line 85, please correct “maded” to “made”

Authors' response: We are thankful for the reviewer’s corrections.

Action: We already correct this grammar mistake

3.Please correct the Ref.51

Authors' response: We are thankful for the reviewer’s corrections.

Action: A correction was made to the original paper, and the reference was updated.